# Assessing Fish Species Tolerance in the Huntai River Basin, China: Biological Traits versus Weighted Averaging Approaches

**Xiao-Ning Wang [1], Hai-Yu Ding [2], Xu-Gang He [1], Yang Dai [3], Yuan Zhang [3] and Sen Ding [3,\*]**

[1] College of Fisheries, Huazhong Agriculture University, Wuhan 430079, China; xiaoninghzau@163.com (X.-N.W.); xgh@mail.hzau.edu.cn (X.-G.H.)
[2] Department of Ecology and Ecosystem Management, Technische Universitaet Muenchen, Freising-Weihenstephan 85354, Germany; haiyu.ding@gmail.com
[3] State key Laboratory of Environmental Criteria and Risk Assessment, Chinese Research Academy of Environment Sciences, Beijing 100012, China; daiyang815@126.com (Y.D.); zhangyuan@craes.org.cn (Y.Z.)
\* Correspondence: dingsen@mail.hzau.edu.cn

**Abstract:** Fish species tolerance used as a component of fish-index of biological integrity (F-IBI) can be problematic as it is usually classified using the historical data, data from literature or expert judgments. In this study, fish assemblages, water quality parameters and physical habitat factors from 206 sampling sites in the Huntai River Basin were analyzed to develop tolerance indicator values (TIVs) of fish based on a ($F_b$-TIVs) and the weighted averaging (WA) method ($F_W$-TIVs). The two quantitative methods for fish tolerance were then compared. The $F_W$-TIVs and $F_b$-TIVs of fish species were calculated separately using a WA inference model based on ten water quality parameters (WT, pH, DO, SC, TDS, $NH_3$, $NO_2^-$, $NO_3^-$, TP, $Cl^-$, and $SO_4^{2-}$), and six biological traits (lithophilic spawning, benthic invertivores, cold water species, equilibrium or periodic life history strategies, families of Cottidae, and species distribution range). Fish species were then classified into biological traits approach three categories (tolerant species, moderately tolerant species, and sensitive species). The results indicated that only 30.3% fish species have the same classification based on $F_W$-TIVs and $F_b$-TIVs. However, the proportion of tolerant species based on two methods had a similar response to environmental stress, and these tolerant species were correlated with PCA axes 1 site scores obtained by ($F_W$-TIVs, $p < 0.05$, $R^2 = 0.434$; $F_b$-TIVs, $p < 0.05$, $R^2 = 0.334$) and not correlated with PCA axis 2 site scores ($F_W$-TIVs, $p > 0.05$, $R^2 = 0.001$; $F_b$-TIVs, $p > 0.05$, $R^2 = 0.012$) and PCA axis 3 site scores ($F_W$-TIVs, $p > 0.05$, $R^2 = 0.000$; $F_b$-TIVs, $p > 0.05$, $R^2 = 0.013$). The results of linear regression analyses indicated that $F_b$-TIVs can be used for the study of fish tolerance. Fish tolerance assessments based on $F_W$-TIVs requires long-term monitoring of fish assemblages and water quality parameters to provide sufficient data for quantitative studies. The $F_b$-TIV method relies on the accurate identification of fish traits by an ichthyologist. The two methods used in this study can provide methodological references for quantitative studies of fish tolerance in other regions, and are of great significance for the development of biological assessment tools.

**Keywords:** fish species tolerance; Huntai river basin; biological traits approach; weighted averaging method; water quality parameters; physical habitat factors

## 1. Introduction

Aquatic organisms are commonly used as biological indicators of river ecosystem health [1] and are used to assess the state of a system relative to its reference conditions. This provides comprehensive information that could be missed or ignored during routine monitoring based on

physicochemical parameters [2]. The Index of Biological Integrity (IBI) established by Karr [3] is a very useful fish community indicator that has been applied to other aquatic life such as algae, macro-benthos and macrophytes [4–6]. Fish IBI (F-IBI) is a multi-indices system, which describes five biological characteristics [3]. One of these characteristics is species tolerance. Tolerance values are generally obtained from historical records, publications or expert judgments from freshwater F-IBI research around the world, including Victoria Lake, Kenya [7], Chybaish marsh, Iraq [8], and the Guem River, Korea [9].

The sensitivity or tolerance of species is an appropriate indicator of aquatic environmental quality [10], since species survive at locations characterized by high oxygen, clean water, good habitat quality, and low levels of human disturbance. On the contrary, these species are poorly distributed at locations with low oxygen, turbid water, high concentrations of pollutants, and intensive human activities. Using fish species tolerance as a measure of F-IBI is problematic, as there is a lack of data for many countries, including China [11]. In most cases, the characteristics of fish tolerance available from the literature and historical data are only available for a small number of species. Thus, if a species tolerance is unknown, it is often ignored when calculating F-IBI, or else classified according to the existing classification for other species in the same genus or expert judgments. However, the degree of sensitivity and capacity of a species to adapt to changing environmental conditions is often species specific [12]. This subjective classification ignores the interspecific differences of tolerance within the same genus. Carlisle et al. reported that two fish species in the same genus (*Etheostoma flabellare* and *E. olmstedi*) sampled from Appalachian streams exhibited interspecific differences [13]. *E. flabellare* was an intolerant species, whereas *E. olmstedi* was a tolerant species. Although fish species may be in the same genus, a temporal small-scale disturbance can separate their spatial niches and change their living environment [14]. This indicates that fish species within a genus may have different variability or environmental tolerance under the same circumstances.

The collection of water quality parameters and biological data during routine water quality monitoring addresses the relationship between fish species and general environmental disturbances [15]. For example, the application of the weighted averaging (WA) method based on available datasets [16–18] can be used to quantify fish tolerance. The wide range of applications of qualitative classification of fish ($F_W$-TIVs) in the United States provides a reliable framework for the development of fish species tolerance indicator values in other regions. Fish tolerance assessment based on $F_W$-TIVs requires a large amount of data on individual fish species and environmental stress factors, which requires long-term aquatic ecology monitoring. However, this data is not currently available for many areas.

Species traits also reflect tolerance to environmental conditions [19]. For example, fish that are lithophilic spawners and benthic invertivores are intolerant to habitat alteration [20], while cold-adapted species are sensitive to increases in water temperature [21], and species that have equilibrium or periodic life history strategies are sensitive to changes in flow regimes [22,23]. Therefore it is important to understand the interaction between various environmental threats and fish reactions, as specific traits are likely to determine fish responses to different environmental threats [24–26]. A combination of species traits and environmental stress provides a new horizon for assessing fish species tolerance, without the need for complicated indoor water chemistry analysis of water quality parameters, and errors of water quality test values associated with seasonal variation (i.e., the water volume increases in the wet season and decreases in the dry season).

The current study aims to quantify fish tolerance using the biological traits approach ($F_b$-TIVs) and WA method ($F_W$-TIVs), and to compare the application of the two methods for fish tolerance assessment in aquatic ecological monitoring. The quantitative results of fish tolerance arising from this study will be of great use in the development of fish diversity tools.

## 2. Methods

### 2.1. Study Area

The Huntai River Basin (40.45°~40.30° N, 122.00°~125.30° E) is located in Liaoning Province, China, and covers an area of $2.73 \times 10^4$ km². The Hun and Taizi rivers rise in the Changbai Mountain Range and converge near the Daliao River, before flowing into the Bohai Sea at Yingkou City, Liaoning Province. The Hun River is 415 km long and Taizi River is 413 km long. The study area has a temperate monsoon climate. The Huntai River Basin is comprised of forested land (46.26%), dry land (22.54%), paddy fields (10.84%), and built urban land (15.45%). The upper and middle reaches of the river basin are mainly covered by deciduous broad-leaved forests, while the lower plain is characterized by intensive human activity and urbanization.

### 2.2. Fish Sampling

Sampling was carried out at a total of 206 sites in August, 2009 and July, 2010 (Figure 1). Fish were collected using electrofishing equipment and gill nets. Two technical operators with professional experience in backpack electrofishing (LR-24; Smith-Root Inc., Vancouver, WA, USA) collected fish along a zig-zag path within a 200-m reach over a 30-min period. Two gill nets with mesh sizes of $3 \times 3$ cm and $6 \times 6$ cm, respectively, were set in the water for 2 h as supplementary fish collection when a reach was not accessible by wading (deeper than 1.5 m). Fish were counted, identified and recorded immediately after collection. All fish were then released, with the exception of that that could not be identified in the field. Unidentified fish were stored in 10% formalin solution and brought back to the laboratory for identification.

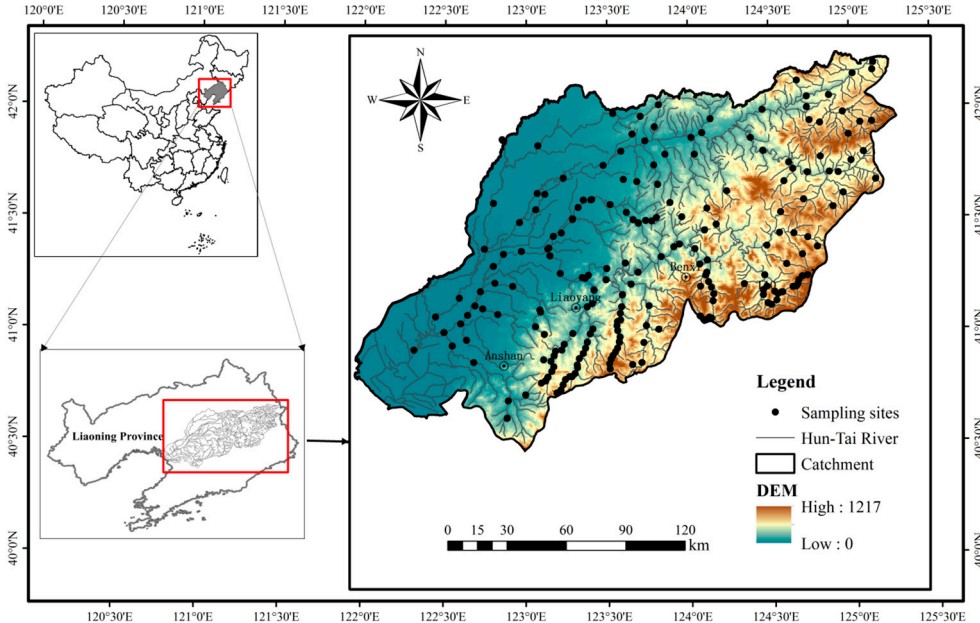

**Figure 1.** Location of 206 sampling sites in the Huntai River Basin. Sampling was carried out at these sites in August 2009 and July 2010.

### 2.3. Water Quality Parameters

Physical and chemical water parameters including water temperature (WT, °C), pH, dissolved oxygen (DO, mg/L), specific conductivity (SC, μS/cm), total dissolved solids (TDS, mg/L), ammonia ($NH_3$, mg/L), nitrites ($NO_2^-$, mg/L), nitrates ($NO_3^-$, mg/L), total phosphorus (TP, mg/L), chloride ions ($Cl^-$, mg/L), and sulfate ($SO_4^{2-}$, mg/L) were recorded. Water quality sampling at each site was carried out 500 m upstream of the fish sampling area to avoid any interference of fish sampling activity

on water quality results. Water quality parameters including WT, pH, DO, SC, and TDS were measured with a portable Water Test Kit (YSI Pro 2030, Yellow Springs, OH, USA) in the field. Three water samples (one from the left bank, one from the middle channel and one from the right channel) were collected mixed into a 2 L bottle which was stored in the freezer. These water samples were transported to the laboratory within 48 h of collection to measure $NH_3$, $NO_2^-$, $NO_3^-$, TP, $Cl^-$, and $SO_4^{2-}$. Water quality parameters were determined following the methods outlined in the environmental quality standards for surface water [27].

### 2.4. Physical Habitat Factors

The quality of physical habitats within the river were determined using the habitat evaluation index system. This system includes ten habitat factors: substrate, habitat complexity, velocity–depth combination, intensity of human activities, riverside land use, bank stability, channel alteration, stream flow conditions, vegetation diversity, and water quality conditions [28,29]. Each parameter has four scoring categories: optimal (score of 16–20), suboptimal (score of 11–15), marginal (score of 6–10), poor (score of 0–5). The score for each parameter at each sampling site was determined by expert visual evaluation in the field (Table A1).

### 2.5. Data Analysis

#### 2.5.1. Calculation of $F_W$-TIVs

$F_W$-TIVs of fish species were calculated using a WA [16,30] inference model based on ten water quality parameters (TW, pH, DO, SC, TDS, $NH_3$, TP, $Cl^-$, $SO_4^{2-}$, and $NO_x$ ($NO_x = NO_2^- + NO_3^-$)) [17]. To increase the confidence of the estimates, species with less than thirty individuals recorded across all sites were removed before calculations [10]. The WA for each species under each water quality parameter was calculated using the following formula:

$$WA_j = \frac{Y_1 X_1 + Y_2 X_2 + \ldots + Y_n X_n}{Y_1 + Y_2 + \ldots + Y_n} \tag{1}$$

where the weighted average for each species *j* is expressed by $WA_j$; *X* is one of the water quality parameters measured from site 1, 2, . . . , *n*; *Y* is the individual number of species *j* from site 1, 2, . . . , *n*. For example, in this study the WA of *Abbottina liaoningensis* for TW was obtained by multiplying the TW value for each site by the number of *A. liaoningensis* recorded at that site. The sum of this value for each site was then divided by the total number of *A. liaoningensis* recorded at all sampling sites.

The WA for each species under each water quality parameter was calculated separately to get a WA data set. The WA for each species was transformed into an ordinal rank using a 10-point ordinal classification [30,31] based on the 10% quantile of WAs. A species $F_W$-TIV was equal to the mean of the ordinal ranks of ten WA, and fish species were classified into three categories according to $F_W$-TIV: tolerant species ($F_W$-TIV = 1–4), moderately tolerant species ($F_W$-TIV = >4–<7), and sensitive species ($F_W$-TIV = 7–10) [17].

#### 2.5.2. Calculation of $F_b$-TIVs

Both tolerance and sensitivity indicate the extent to which a species is physically or behaviorally affected by environmental conditions. Based on fish species traits and the sensitivity association approach [19,20], six biological traits (lithophilic spawning, benthic invertivores, cold water species, equilibrium or periodic life history strategies, families of Cottidae, and species distribution range) were selected to describe fish species tolerance. Species were also classified according to key biological traits [32], and each trait except for species range distribution was assigned a score of 0 (species do not have this trait) or 1 (species have this trait). The score for fish range distribution was calculated as the frequency of occurrence subtracted from one. The scores were scaled from 0 for species found in all sites to 1 for species found in a single site, with the exception of the minimum and maximum

score. The frequency of occurrence for each species was given a score within this scale (e.g., a species found in 35% of sites would be given a score of 0.65, while one found at 75% of sites would be given a score of 0.25). The above scoring rules followed those described by Sievert et al. [20] and the $F_b$-TIVs were the sum of all six traits. For comparison with $F_W$-TIV values, $F_b$-TIV values were also divided into three groups, i.e., tolerant species ($F_b$-TIV = 0–2), moderately tolerant species ($F_b$-TIV = >2–<4), sensitive species ($F_b$-TIV = 4–6).

### 2.5.3. Statistical Analysis

In order to compare the response of tolerant species to environmental disturbances based on the two methods, major environmental pressure gradients were established. The Shapiro–Wilk test was used to determine whether the data were normally distributed, and Spearman's correlation coefficient was then used to remove the redundant parameters (|Pearson's coefficients| > 0.7) [30]. Principal component analysis (PCA) was applied to identify the major environmental pressure gradients [33]. The number of PCA axes were determined using Kaiser's rule requiring eigenvalues >1. Parameters were considered as major stressors, when the absolute value of loading for axes was ≥0.5. One-way analysis of variance (ANOVA) was used to analyze the differences in fish tolerance classifications based on two methods. To assess the concordance of tolerant species based on $F_w$-TIV values and $F_b$-TIV values to environment stress, linear regression was used. Data were transformed (lg(x+1)) before analysis. Correlation analysis and PCA were performed using CANOCO 4.5. One-way ANOVA was performed using SPSS 19.0 and linear regression was conducted using Sigmaplot 12.5.

## 3. Results

### 3.1. The Relationship between the $F_W$-TIV Method and the $F_b$-TIV Method for Species Tolerance Classification

A total of 33,102 fish representing 42 species were collected from 206 sampling sites. Prior $F_W$-TIV and $F_b$-TIV calculations, nine species that had a low frequency of occurrence were excluded from the dataset. Therefore, calculations were carried out using data relating to 33,013 individuals representing 33 species.

$F_W$-TIV results found that seven species (21.2%) were tolerant, 18 species (54.6%) were moderately tolerant and eight species (24.2%) were sensitive. *Cottus poecilopus* ($F_W$-TIV = 10.0) and *Hypomesus olidus* ($F_W$-TIV = 9.1) were the most sensitive species. The two most tolerant species were *Oryzias latipes* ($F_W$-TIV = 1.9) and *Hemiculter leucisculus* ($F_W$-TIV = 2.0). However, fish tolerance classification based on $F_b$-TIV values found that 26 species (78.8%) were tolerant, six species (18.2%) were moderately tolerant and a single species (3%) was sensitive. *C. poecilopus* ($F_b$-TIV = 4) was the most sensitive species, while *Cobitis granoei* ($F_b$-TIV = 0.6) and *Misgurnus anguillicaudatus* ($F_b$-TIV = 0.6) were the most tolerant species (Table A2). One-way ANOVA revealed that there was a significant different between fish tolerance classification based on the two methods (F = 28.5, P = 0).

### 3.2. The Relationship between $F_w$-TIV, $F_b$-TIV and Environmental Pressure Gradients

Pearson's correlation coefficients showed high redundancy between SC and TDS (r = 0.794, p < 0.01), SC and Cl$^-$ (r = 0.822, p < 0.01), SC and SO$_4{}^{2-}$ (r = 0.828, p < 0.01), Cl$^-$ with SO$_4{}^{2-}$ (r = 0.724, p < 0.01), Cl$^-$ with NH$_3$ (r = 0.789, p < 0.01), TDS with Cl$^-$ (r = 0.705, p < 0.01), and TDS with SO$_4{}^{2-}$ (r = 0.907, p < 0.01). In this study, TDS, SC and Cl$^-$ were removed, whereas SO$_4{}^{2-}$ was retained for further analysis as sulfate is a dominant ion reflecting background characteristics of ion composition in the mountainous river.

Four significant PCA axes were derived, which explained 67.8% of the variation among these environmental parameters. Axis 1 was most heavily influenced by five physical habitat factors (substrate, habitat complexity, velocity-depth combination, water quality conditions, intensity of human activities) and four water quality parameters (WT, DO, SO$_4{}^{2-}$, NH$_3$), representing 39.9% of the variation. Axis 2 was most strongly associated with vegetation diversity which accounted for 10.3%

of the variations. Axis 3 explained 10% of the variation and included $NO_x$. Axis 4 only explained 7.6% of the variation and included intensity of human activities and riverside land use (Table 1). As PCA axes 1, 2, and 3 explained most of the data variability, they were used as stressor gradients for subsequent analysis.

**Table 1.** Eigenvector matrix for principal component analysis (bold values are considered high $\geq |0.5|$). WT: water temperature; DO: dissolved oxygen; TP: total phosphorus.

| Items | Axi1 | Axi2 | Axi3 | Axi4 |
|---|---|---|---|---|
| Substrate | **−0.8569** | 0.0562 | 0.0997 | 0.2706 |
| Habitat complexity | **−0.7431** | 0.2754 | 0.1846 | 0.2906 |
| Velocity-depth combination | **−0.7194** | 0.2810 | 0.0990 | 0.2664 |
| Bank stability | −0.4523 | 0.4871 | −0.0140 | −0.0551 |
| Channel alteration | −0.3885 | 0.2352 | 0.0617 | −0.2449 |
| Stream flow conditions | −0.3481 | −0.0686 | −0.4399 | −0.2449 |
| Vegetation diversity | −0.2241 | **0.6394** | 0.2703 | −0.3494 |
| Water quality conditions | **−0.7110** | −0.1939 | 0.3191 | −0.2219 |
| Intensity of human activities | **−0.5963** | −0.0299 | 0.1136 | **−0.5523** |
| Riverside land use | −0.2955 | 0.2746 | 0.2010 | **−0.6377** |
| WT | **0.6404** | 0.0177 | 0.3955 | −0.0094 |
| pH | −0.3131 | −0.3660 | −0.2698 | −0.0087 |
| DO | **−0.6137** | −0.4973 | −0.1304 | −0.1228 |
| TP | 0.4629 | 0.3555 | −0.3085 | −0.0092 |
| $SO_4^{2-}$ | **0.7642** | 0.2781 | 0.4459 | 0.1492 |
| $NO_X$ | 0.2364 | −0.4163 | **0.7136** | −0.0329 |
| $NH_3$ | **0.6945** | 0.3964 | −0.2868 | 0.0299 |

Proportion of tolerant species obtained by $F_W$-TIV correlated with PCA axis 1 site scores ($p < 0.05$, $R^2 = 0.434$; Figure 2a), but had no correlation with PCA axis 2 site scores ($p > 0.05$, $R^2 = 0.001$; Figure 2c) and PCA axis 3 site scores ($p > 0.05$, $R^2 = 0.000$; Figure 2e). Proportion of tolerant species obtained by $F_b$-TIV correlated with PCA axis 1 site scores ($p < 0.05$, $R^2 = 0.334$; Figure 2b), although there was no correlation with PCA axis 2 site scores ($p > 0.05$, $R^2 = 0.012$; Figure 2d) and PCA axis 3 site scores ($p > 0.05$, $R^2 = 0.013$; Figure 2f).

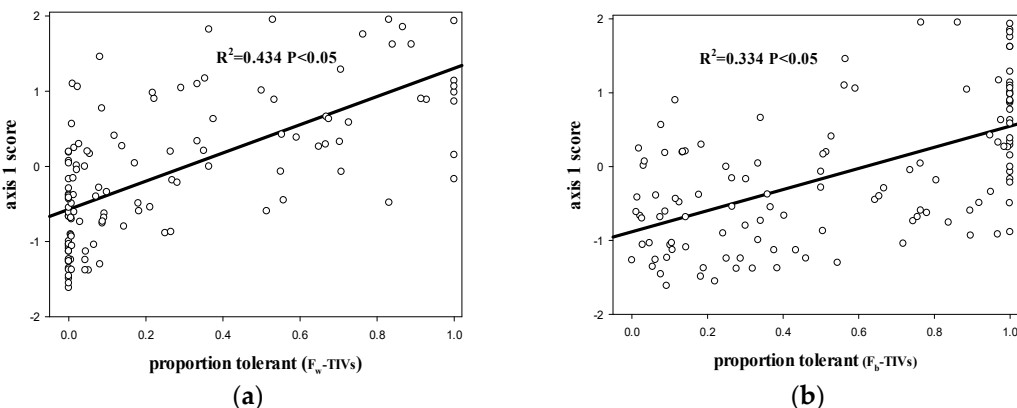

**Figure 2.** *Cont.*

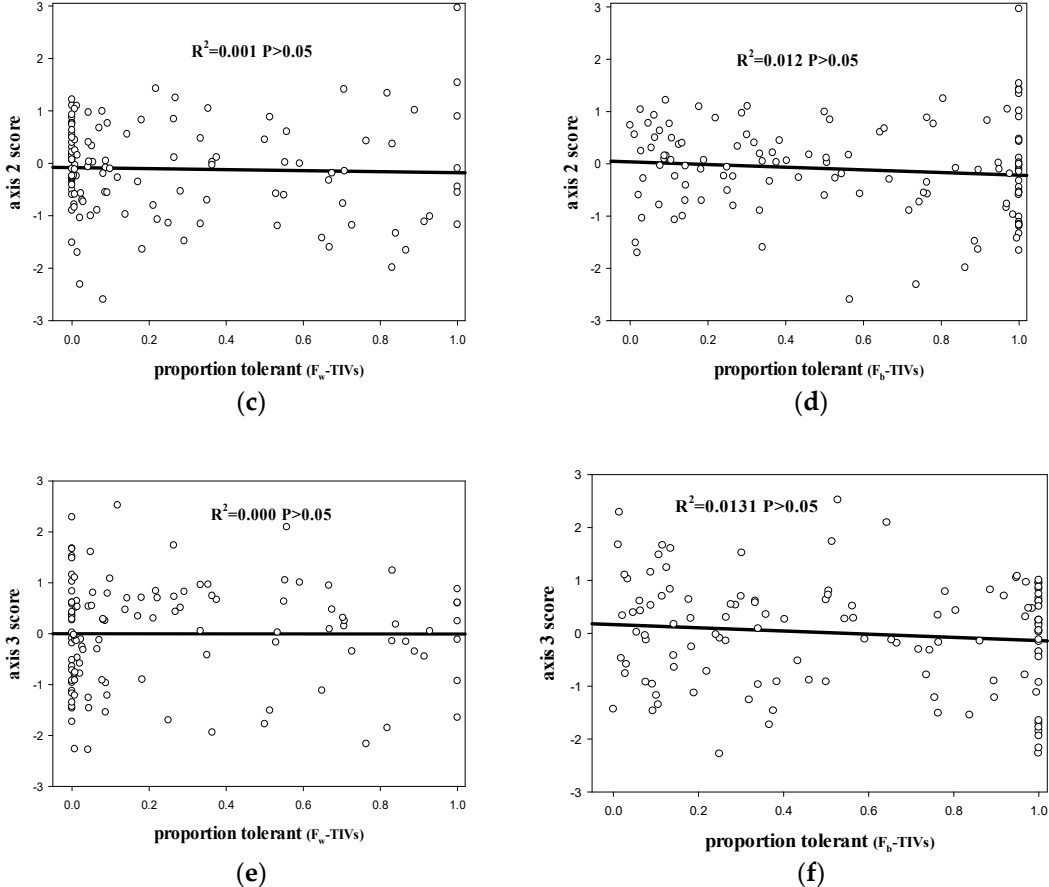

**Figure 2.** (**a**) scatterplot of PC1 site scores versus the proportion of tolerant fish at each site based on F$_W$-TIVs; (**b**) scatterplot of PC1 site scores versus the proportion of tolerant fish at each site based on F$_b$-TIVs; (**c**) scatterplot of PC2 site scores versus the proportion of tolerant fish at each site based on F$_W$-TIVs; (**d**) scatterplot of PC2 site scores versus the proportion of tolerant fish at each site based on F$_b$-TIVs; (**e**) scatterplot of PC3 site scores versus the proportion of tolerant fish at each site based on F$_W$-TIVs; (**f**) scatterplot of PC3 site scores versus the proportion of tolerant fish at each site based on F$_b$-TIVs. Line is best fit. (F$_b$-TIVs: tolerance indicator values (TIVs) of fish based on biological traits approach. F$_W$-TIVs: tolerance indicator values (TIVs) of fish based on weighted averaging (WA) method).

## 4. Discussion

Determining fish tolerance based on quantitative methods (F$_w$-TIV and F$_b$-TIV) is more advantageous than using qualitative methods (e.g., expert judgments) in the application and development of biological assessment tools. Quantitative methods are better able to identify fish tolerance characteristics on environmental stress. Segurado et al. [10] suggested that experts tend to classify species which occur less frequently as moderately tolerant species. Species that were recorded to occur less frequently in the current study, such as *Pseudogobio vaillanti* (present at 5 of 206 sites) and *Oryzias latipes sinensis* (present at 17 of 206 sites), were considered moderately tolerant species by experts [34]. Both methods used in the current study found that these species were tolerant. Tolerance as defined by Bressler et al., is the ability of a living organism to survive and reproduce under environmental stress [35]. Tolerance classifications based on expert opinions are vague and may result in an underestimation of the tolerance of some fish species. For example, *Phoxinus lagowskii* and *Opsariichthys bidens* were classified as sensitive species by experts due to their distribution [34], however both methods used in the current study demonstrated that they were moderately tolerant species. Species distribution may result from historical factors rather than species sensitivity to environmental stressors. Quantitative methods may be more suitable for the development of a new biological

assessment index. For example, the development of Biotic Indices (BI) and the Biological Monitoring Working Party (BMWP) are based on quantitative tolerance values of benthic macro-invertebrates [36].

The fish tolerance assessment system developed in the current study is based on the fact that traits are the result of environmental adaptation [37]. For example, *Carassius auratus* and *M. anguillicaudatus* are classified as tolerant species based on $F_b$-TIV values. Both of these species can be found in many types of water bodies in China, especially in urban reaches with low oxygen and high pollution levels. Their strong dispersal ability helps them adept to environmental changes, as they can choose favorable habitats for survival [38]. Furthermore, they are omnivores, which means they can consume a wide variety of prey, aiding their survival [39,40] when inhabiting highly disturbed environments. The relationship between fish biological traits and sensitivity also has been recognized by other researchers. Hermoso et al. [41] found the widely distributed species *Luciobarbus comizo* showed a higher tolerance than species that are only found upstream in Guadiana River basin. Species with low sensitivity have a relatively high potential for species distribution under environmental changes (i.e., pH, salinity, temperature) [42]. The biological traits approach helps to explain the adaptation mechanisms of fish in specific environmental conditions [43] and examines fish traits in relation to environmental pressure gradients as a novel approach in assessing fish tolerance.

It must be noted, however, that fish tolerance classification based on $F_b$-TIV showed a significant difference when compared with $F_w$-TIV. These discrepancies may be due to the number of available traits. In order to improve the accuracy of the assessment results, more traits would need to be included. It is well known that the fish traits such as morphology and body size vary in the process of environmental adaptation [44,45]. Generally, fish with spindle-shaped bodies display strong swimming ability to evade the impact of environmental pressures, and larger-bodied species can more readily populate new regions to avoid substrate changes due to the positive correlation of body size with home range size [46]. In addition, the reproductive period of fish is considered to be sensitive to changes in river flow [47]. These theories suggest that it is important to provide multifarious traits to improve the efficacy of the biological traits approach. Secondly, fish traits were assigned 0 or 1 scores in this study, which may limit the results of fish tolerance classification. If more grade divisions are allocated for each trait, the classification of a fish species tolerance may change.

The stressors affecting fish tolerance in the $F_w$-TIV method have been the focus of previous research. Fedorenkova et al. [48] stated that physical parameters should be considered in species tolerance studies. With the degraded gradient of riparian vegetation, sensitive fish species are gradually disappearing and being replaced by tolerant species [49]. Casatti et al. [50] found that the abundance of fish species changed with physical habitat index, and some species showed optimal distribution corresponding to the degree of habitat conservation. The PCA analysis results from the current study also indicate that physical habitat factors are important environmental stress factors. However, measured physical habitat factors were not used to calculate $F_w$-TIV values in the current study. There are many environmental factors affecting fish, including natural variables such as elevation, slopes and longitude, anthropogenic variables such as hydro-morphology, land use, and substrate. However, water quality is the most direct factor affecting fish. Therefore, the $F_w$-TIV method used in this study is a reliable means of identifying fish tolerance. In order to improve the accuracy of the $F_w$-TIV method, more environmental factors will be included in future studies.

Geographical scales should also be recognized as a common problem between the two methods used. Many researchers have found that the tolerance range of individual species varies between different regions [30,51]. From the individual level of fish, the geographical distribution of river basins determines the broader environmental gradient that ultimately selects suitable species traits [52]. Lourenco et al. [53] found that the maturation time of *Piabina argentea* differed between basins. Fish living in different eco-regions are subject to different levels of environmental stresses. Some researchers have indicated that fish reproduction can be promoted or inhibited by different levels of environmental stress [54,55]. The fish tolerance values obtained in the current study are for an eco-region with similar aquatic ecosystems. For example, Whittier and Hughes [56] divided their study area into three

eco-regions to calculate the tolerance values of fish and amphibians. The United States Environmental Protection Agency have developed five regional systems for TIVs of macro-invertebrates [29] in order to reduce estimation errors caused by regional differences in environment.

Both methods used in the current study can be used for quantitative studies of fish tolerance and both have advantages in application. For the $F_w$-TIV method, data from a high number of individuals and environmental stress factors from long-term aquatic ecology monitoring are required. However, the development of emerging technologies and informatization makes processing $F_w$-TIV values simple. If there is a large amount of biological monitoring available data for an eco-region, it is more convenient to use the $F_w$-TIV method for identifying fish tolerance. For example, fish communities and water quality are monitored in U.S. for the long-term National Water-Quality Assessment Program and the collected data can be used directly by managers for $F_w$-TIV analysis. In the $F_b$-TIV method, less survey effort is required, but professional identification of fish traits by an ichthyologist is needed. When aquatic ecology data is not available in some eco-regions, the $F_b$-TIV method is acceptable for the quantitative study of fish tolerance in these areas. The two methods used in the current study provide methodological references for quantitative studies of fish tolerance in other regions and are of great significance for the development of biological assessment tools.

**Author Contributions:** All authors listed have contributed to this study. X.-N.W., S.D. and G.-H.X. had a substantial involvement in the conception, guidance, and revising of the manuscript. The data acquisition and analysis were done by H.-Y.D., H.D. and Y.Z. The manuscript was written by X-N.W. and S.D.

**Funding:** This study was supported by grants from the National Natural Science Foundation of China (41401066), the National Natural Science Foundation of China (41571050), and the Fundamental Research Funds for the Central Universities (2662015QD004).

**Acknowledgments:** All authors listed have contributed to this study. Xianwei Huang supplied excellent field assistance.

**Conflicts of Interest:** The authors declare no conflict of interest.

## Appendix A

**Table A1.** Habitat assessment scoring descriptions.

| Factor | Optimal | Suboptimal | Marginal | Poor |
|---|---|---|---|---|
| Substrate | More than 75% composition of gravel, cobbles and big stones | 50%–75% composition of gravel, cobbles and big stones | 25%–50% composition of gravel, cobbles and big stones | Less than 25% composition of gravel, cobbles and big stones |
| Habitat complexity | Composition of aquatic vegetation, litter, fallen wood, concave banks and boulders, etc. | Composition of aquatic vegetation, litter, fallen wood, and concave banks, etc. | Domination by one or two kinds of microhabitat | Domination by one kind of microhabitat and the substrate mainly composed by silt or fine sand |
| Velocity-depth combination | Slow (<0.3 m/s)-deep (>0.5 m); slow-shallow (<0.5 m); fast (>0.3 m/s)-deep; fast-shallow three types of habitats | Only three types of habitat (fast-shallow type got the highest value) | Only two types of habitat (in the absence of fast-shallow type and slow-shallow type) | Predominated by one velocity–depth type (usually pools) |
| Intensity of human activities | Hardly any human disturbance | Minimal human disturbance by few walkers or bikes | Less human disturbance by vehicles | Serious human disturbance by motor vehicles |
| Riverside land use | No agricultural land on either side of the river bank | Agricultural land present on one side of the river bank | Agricultural land present on both sides of the river bank | Weathered soils after fallow conditions present on both sides of the river bank |
| Bank stability | No erosion on the river banks. Less than 5% of the bank is damaged in the visual range (100 m) | 5%–30% erosion of the bank in the visual range (100 m) | 30%–60% erosion of the bank in the visual range (100 m) | More than 60% erosion of the bank in the visual range (100 m) |

**Table A1.** *Cont.*

| Factor | Optimal | Suboptimal | Marginal | Poor |
|---|---|---|---|---|
| Channel alteration | No channelization of the river | Less channelization around the pier | Embankments or bridge pillars on both sides of the strait and more extensive channelization of the river | River bank fixed by wire and cement |
| Stream flow conditions | Large water volume, only a few exposed areas of the bank visible | Relatively large volume and 75% of the river bank is covered | Relatively high volume covering 25%–75% of the river bank | Small water volume and dry river course |
| Vegetation diversity | Over 50% coverage of vegetation | 25%–50% vegetation coverage | Less than 25% vegetation coverage | Hardly any vegetation coverage |
| Water quality conditions | Low turbidity, no sedimentation, no odor from the river water | Low turbidity, a small amount of odor from the river water | Water with a high turbidity and odorous water | High turbidity and foul smelling water |
| score | 15–20 | 10–15 | 5–10 | 0–5 |

Note: reference as [28].

**Table A2.** Summary of $F_w$-TIVs and $F_b$-TIVs for fish species and tolerance classifications.

| Species | $F_w$-TIVs | $F_b$-TIVs |
|---|---|---|
| *Abbottina liaoningensis* | 6.3 (M) | 1.8 (T) |
| *Abbottina rivularis* | 7.1 (S) | 1.6 (T) |
| *Cobitis granoei* | 6.7 (M) | 0.6 (T) |
| *Barbatula barbatula nuda* | 4.3 (M) | 2.4 (M) |
| *Rhodeus lighti* | 4.2 (M) | 0.9 (T) |
| *Rhodeus sinensis* | 4.6 (M) | 1.0 (T) |
| *Squalidus chankaensis* | 5.6 (M) | 2.0 (T) |
| *Squalidus wolterstorffi* | 7.6 (S) | 1.0 (T) |
| *Leuciscus waleckii* | 8.3 (S) | 2.9 (M) |
| *Carassius auratus* | 4.1 (M) | 0.6 (T) |
| *Zacco platypus* | 6.7 (M) | 1.6 (T) |
| *Rostrogobio liaohensis* | 5.9 (M) | 1.0 (T) |
| *Gobio lingyuanensis* | 3.4 (T) | 2.0 (T) |
| *Gobio rivuloides* | 2.1 (T) | 1.0 (T) |
| *Gobio cynocephalus* | 4.1 (M) | 2.9 (M) |
| *Phoxinus lagowskii* | 6.2 (M) | 2.4 (M) |
| *Opsariichthys bidens* | 4.7 (M) | 2.0 (T) |
| *Pseudorasbora parva* | 5.6 (M) | 1.6 (T) |
| *Hemiculter leucisculus* | 2.0 (T) | 0.9 (T) |
| *Huigobio chinssuensis* | 8.2 (S) | 0.9 (T) |
| *Pseudogobio vaillanti* | 2.3 (T) | 2.0 (T) |
| *Acheilognathus chankaensis* | 2.4 (T) | 0.9 (T) |
| *Misgurnus anguillicaudatus* | 5.0 (M) | 0.6 (T) |
| *Lefua costata* | 4.5 (M) | 0.8 (T) |
| *Hypomesus olidus* | 9.1 (S) | 3.0 (M) |
| *Lampetra mori* | 6.6 (M) | 2.0 (T) |
| *Perccottus glenni* | 5.1 (M) | 1.0 (T) |
| *Ctenogolius brunneus* | 4.0 (T) | 2.8 (M) |
| *Cottus poecilopus* | 10.0 (S) | 4.0 (S) |
| *Hypseleotris swinhonis* | 4.6 (M) | 0.9 (T) |
| *Odontobutis yaluensis* | 7.8 (S) | 1.8 (T) |
| *Oryzias latipes sinensis* | 1.9 (T) | 0.9 (T) |
| *Pungitius pungitius* | 9.0 (S) | 0.9 (T) |

Note: T represents tolerant species, M represents moderate species, S represents sensitive species.

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
