# Peer review of "Assessing Fish Species Tolerance in the Huntai River Basin, China: Biological Traits versus Weighted Averaging Approaches"

_water, doi:10.3390/w10121843_

Round 1

Reviewer 1 Report

GENERAL COMMENTS

This manuscript focuses on an interesting subject, which is the determination of fish species tolerance (so often assessed only through expert judgement) which are useful, for example in multi-metric ecological quality indices. The manuscript, however, present several important concerns that must be dealt first, before a decision on acceptance can be made. First of all, the English language, which is very poor, and in some sections, difficult to read or interpret. If they want to proceed with the revision, it is imperative that this manuscript must be revised by a scientific native English speaker. There are also a lot of information that lack details and the authors commonly employ vague expression to explain their methodologies. For example, the authors, besides water quality paraments, the authors assessed other natural environmental variables and also pressure variables. But most of them (if not all), lacks explanation on how they were measured and assessed. Therefore, the authors MUST explain how each one of these variables was measured (field procedures) and assessed (for example “substrate” – how many classes were considered? All these variables should be called by the same name as they appear in the other parts of the manuscript (see comments below). The authors assessed 2 methods to determine species tolerance and they say BTIV is better than WTIV. But according to the results outlined, it is difficult to understand why BTIV performed better, as I don’t see any test comparing both approaches. It seems also, authors doubt about the validity of their own findings as on line 248 they say “the listed traits may not provide enough information to identify fish tolerance”. Further, on line 258, they say “we provided fewer biological traits information for fish tolerance assessment”. So If the listed traits are not adequate or insufficient, I guess the findings of your study are biased. Is that so? I would like the authors to comment these issues. The concluding paragraph (and also the last sentence of the Abstract) was quite disappointing, as readers would like to read answers to these questions: i) Of the 2 methods tested, what is the most appropriate (BTIV or WTIV?); ii) What are the implications of your study to future studies? Iii) Can your findings be applicable to other contexts, other than the present one in China? Please assess this on your revised version.

Specific comments are outlined below and include other issues that should be dealt with, other than these general comments.

The manuscript needs therefore a 2nd round of revisions, incorporating all these points, before a decision can be made.

SPECIFIC COMMENTS

Line 15 – provide full name upon first citation of: BTIV, WA and WTIV.

Line 15 – “There…” - Which region/area/river? Species? (this should be move to line 14). I would suggest to remove “There”.

Line 17 – Why only “tolerant fish”? What about other medium-tolerant fish or intolerant fish (see for example Segurado et al. 2011)?

Line 19 – Replace “axis” by “axes”.

Line 20 – “and physical habitat conditions was an important kind of environmental stress.”. How does this relate to the PCA results? Was this a gradient too? Be clearer.

Line 22 – “WTIV and BTIV respectively.”. The authors should explain briefly above, in what it consists each method.

Line 24 – What does this PCA axis 1 explains (i.e. it is a gradient of what?)

Line 25 – Do not start a sentence with “And”, you can merge both sentence in only one. What do you mean by “less correlation”? They were not correlated at all (P> 0.05), correct?

Line 27-28 – “for developing countries may due to the lack of basic data”. These sentence is unclear, please re-write clearer.

Line 27-28 – “biological traits approach is more suitable”. According to the results outlined above, what makes you think that BTIV performed better than WTIV? I don’t see any comparative metrics between both approaches.

Line 52-53: “adapt to exposure”. This is not clear and deserves clarification.

Line 67-68 – This sentence seems to make few sense here and there is not a clear link with the sentence above (species tolerance values). Could you make this link clearer?

Line 70 –benthic invertivorous fish, not benthic invertivores of fish.

Line 70 – Sentence not clear, please re-write.

Line 72 – “species that have equilibrium or periodic life history strategies are sensitive…”. What do you mean by species that have equilibrium or periodic life history strategies?

Line 75 – You mean “Association OF species”, correct?

Line 76 – Why indoor? These water quality are not supposed to be conducted in the field (i.e. outdoor)?

Line 77 – I do not understand what you mean by “more seasonal stability”. Please clarify.

Line 79 – Replace “a study area” by “the study area”.

Line 80 – Again, what do you mean by BTIV or WTIV?

Line 79-81 – But why the species of this river basin are important? The authors should address the importance of the study they are about to conduct. Do these species have a high conservation value?

Line 81 – Remove “respectively”.

Line 81-82 – Is this not already included, and part of the data treatment, on the first goal (line 79-80)?

Line 82 – This should end with a sentence such as “The results obtained in this study will allow (or are important for)…”.

Line 87 – What do you mean by “is converged”?

Line 89-90 – “69% of total area consists of mountains, 6.1% of hills, and 24.9% of plains.”. This is too vague (how do you define what is a hill or a plain or a mountain?). It would be more interesting to know the proportion of e.g. forests, agriculture (intensive/extensive), urban areas, etc. What about the river network, i.e. tributaries?

Line 94 – Please specify.

Figure 1 – Please highlight the course of the Huntai river, as it is hardly discernable. Also Huntai (as in line 85) or Hun-Tai (as it is here)?

Line 93 – “August in 2009 to July in 2010”. But on line 169 you say “during Aug. 2009 and Jul. 2010”. So 2 sampling campaigns, correct?

Line 97 – “206 sampling sites”. But on line 93 you say 224 sampling sites. Please correct accordingly.

Line 98 – Provide brand and model for the electrofishing equipment. For gill nets, provide number, mesh size, dimensions, etc.

Line 101 – What was the period of the day that nets were placed and lifted? Dusk? Night? This info is needed.

Line 104 – formalin 10%? Other? Please specify.

Line 106-107 – “The location of the water quality parameters at each site was selected 500-m upstream of the fish sampling area to avoid the interference of sampling activities on water quality determination.” This makes no sense. Why not making the analyses of water quality parameters after fish sampling?

Line 113 – “low-temperature incubator.” Temperature?

Line 117-118 – This sentence is unclear, please re-write. Please also detail which physical habitat conditions were assessed and how each one was measured. These details are important.

Line 117-121- There is a lot of confusion here, as the authors mix different types of variables. I suggest you group variables by type, i.e. natural environmental variables (substrate, depth, water velocity,…) and pressure variables (I do not understand what you mean by “intensity of human activities”). The authors MUST explain how each one of these variables was measured (field procedures) and assessed (for example “substrate” – how many classes were considered? Based on a Wentworth scale (Bovee, 1986)?). All these variables should be called by the same name as they appear in the other parts of the manuscript (see comments below).

Bovee, K. D., 1986. Development and evaluation of habitat suitability criteria for use in the instream flow incremental methodology, Fort Collins, CO, U.S. Fish and Wildlife Service Biological Report 86(7).

Line 118-119 – “speed and depth”. Is this one variable (???) or two variables, i.e. “speed, depth…”. Also on line 192, you say something like “velocity-depth combination”. So are these the same? 2 variables or 1 single variable (in this case, explain you do mean by a velocity-depth” combination and what does it mean in practice?)? Please also pay attention throughout the manuscript on how do you call the same variables (in this case, speed (line 118) or velocity (line 192)??)

Line 120 – But what does it mean a value of 0 or 20? What is the significance of this scale for each of the variables?

Line 120 – “river water status,…and water quality)”. See my comment from line 118-120: Could you please explain what does it mean (i.e. how it was defined, measured, assessed) a variable named “river water status” or “water quality”?

Line 138 – So a score of 4, means tolerant (1-4) or moderately tolerant (4-7)? Please correct accordingly.

Line 144-145 – Please give more details what you mean by “equilibrium or periodic life history strategists”, and “species distribution range” (what values does this metric assume?).

Line 145-146 – “Each trait … assigned 0 or 1 scores based on fish biological traits [31].” This does not sound well. Again, re-write clear and please provide details.

Line 147 –“percentage of occurrence,”. You mean frequency of occurrence (%) of a given species?

Line 147-148 – “0 for species found in all sites”. I sincerely do not understand why a species found in all sites, receives a 0 (%?) score. Please explain.

Line 152-153- How were these classes defined? Any reference to support this?

Line 157 – Replace “should be” by “were”.

Line 157-159 – “Shapiro-Wilk test determines whether data were normally distributed, if p<0.05, the redundant parameters were removed when they had high correlation with other parameters (|Pearson coefficients|>0.7) [29].”. There seems to be two subjects here. Suggest breaking the sentence in 2: “Shapiro-Wilk test determined whether data were normally distributed. Redundant parameters were removed when they had high correlation with other parameters (|Pearson coefficients|>0.7) [29].”

Line 162 – How only “physicochemical variable”. What about the other variables considered in CCA (i.e. velocity, depth, substrate, etc.)?

Line 163- Suggest instead “was THEN used”.

Line 172-173 – “For most of the species, fish species are moderate based on WTIV, while those obtained by BTIV are tolerant.” How most? There is a lack of details here? How many (or %) as sensitive/moderate/tolerant according to each of those methods?

Line 174 – But what obvious differences? Where are the outputs (table? Figure?) that support these results?

Line 175 – Remove “that”.

Line 177 – “who preferred to live in low temperature and clear waters”. How do you know this (which results support this)? Please remove this part of the sentence.

Line 179 – Replace “who” by “which”.

Line 179-180 – “preferring to inhabit the upper layers of water”. Same comment as on line 177 (remove please).

Line 181- Give also overall picture for the BTIV as you did for the WTIV (lines 175-176).

Line 181 – What do you mean by qualitative and quantitative BTIV and WTIV, respectively? Other? Please place it parenthesis after “qualitative” and “quantitative”. Call things by their names.

Line 192 – “water quality conditions”. I don’t see any variable named “water quality conditions”. Is the same that you named (on line 120) as “water quality”. But what does it mean? Is this a “composed” variable of different parameters?? You need to explain all this on M&M (see my comments below). 

Line 198 – what not using axis 4 as it was selected to be retained (i.e. eigenvalue >1)?

Line 207 – Give complete captions (state the full meaning of your acronyms, refer to the study area name, river,etc.)

Line 214-220 – Suggest instead something like “Figure 3 (a) scatterplot of PC1 site scores versus (a) the proportion of tolerant fish at each site based on WTIV; (b) scatterplot of PC1 site scores versus the proportion of tolerant fish at each site based on BTIV; (c) scatterplot of PC2 site scores versus the proportion of tolerant fish at each site based on WTIV; (d) scatterplot of PC2 site scores versus the proportion of tolerant fish at each site based on BTIV; (e) scatterplot of PC3 site scores versus the proportion of tolerant fish at each site based on WTIV; (f) scatterplot of PC3 site scores versus the proportion of tolerant fish at each site based on BTIV. Line is best fit.”

Line 222 – Figures/tables should not be called on the Discussion.

Line 222-230 – This paragraph is difficult to understand. As in many other section it should be written in clear, concise and objective English, avoiding any source of useless information. Please remove “tolerance to what” and use something more appropriate from the scientific point of view.

“For example, Phoxinus lagowskii and Opsariichthys bidens generally less disturbed”. You mean the species or the sites?

Line 235 – adept or adapt.

Line 237 – “to have more options to receive enough energy”. I do not understand the context of this sentence, particularly when you refer to “energy”.

Line 246-247 – “based on traits method showed a significant difference with WA method”. Please call the method by their names, as you named it in M&M (BTIV, WTIV).

Line 247 – “firstly is due”. I would say, “can be due”.

Line 248 – “the listed traits may not provide enough information to identify fish tolerance”. If the listed traits are not adequate, so I guess the findings of your study are biased. Is that so?

Line 250 – avoid “etc.”

Line 252-253 – “to evade the impact of environmental pressures, and larger-bodied species could more readily populate new regions to avoid some environmental changes [43].”. I don’t know what you mean by “evade the impact of environmental changes”. This needs clarification. Also, specify what you mean by “some environmental changes”.

Line 253 –255 - Please re-write this sentence. You say “reproductive biology traits ---will support more for biological traits approach than life history strategy”. Does this make any sense?

Line 258 – “we provided fewer biological traits information for fish tolerance assessment”. So again, do you think your findings are biased?

Line 267 – “physical habitat conditions is a kind of environmental stressor”. Remove or edit this sentence: physical habitat condition, do not represent an environmental stressor per se.

Line 269-272 – “environmental factors affecting fish species were so much from the ecological point of view, that it is difficult to establish a perfect and accurate fish tolerance assessment system. Water quality parameters as the most direct stress affecting fish communities is still a reliable means to identify fish tolerance.”. But considering water quality alone is insufficient. Besides water quality, natural environmental variables (or gradients) along with human-induced pressures acting on river connectivity, hydrology, morphology and water quality, should be employed for a proper assessment (see for example Segurado et al., 2011).

Line 278 – Please provide the complete context (i.e. where, to what purpose,…).

Line 285 – Why only “developing countries”?

Line 284-293- This last paragraph was disappointing. You cannot only disregard scientifically-valid assessments methods, just because they are expensive. So what is the conclusion of your study? Of the 2 methods tested, what is the most appropriate (BTIV or WTIV?) And most importantly, what are the implications of your study to future studies? Can your findings be applicable to other contexts, other than the present one in China? This is what the readers want to read.

Line 306-307 – There are far more than 8 variables here and not all are physico-chemical. Please correct accordingly.

Line 307 – “bold values are considered high ≥ |0.4|).”. Bold values are all greater than |0.5| as you say on line 162. Pay attention and be consistent!

Author Response

Dear reviewer:

Thanks for your careful review and the valuable suggestions for this article.

Reviewer 2 Report

- Well-written introduction, with good explanation of the background and the study question. the Statistical Analysis section needs some clarification of the different parts.

- Lines 69-78: How does the BTIV relate to the Species Response Index and Trait Association Index of Sievert et al. (2016)?

- Lines 81-82: I'm not sure that I understand this second aim. Is this to test whether there is a relationship between your TIVs and the environmental pressure axes from PCA? In lines 224-225 you state "tolerances to what". Please clarify that this is the aim.

- Line 104: Does this mean released alive?

- Line 141-143: Please add a bit more explanation as to the Sievert et al. (2016) Species Response Index and Trait Association Index approaches, and how they compare to your  current implementation. I think your BTIV is the trait association index? Text can be added here or in introduction (see above - e.g., Line 69).

- Lines 162-164: Does this refer to the regressions between WTIV and BTIV and the PCA axes? It looks like these relationships are still non-linear (e.g., Figure 3a).

- Lines 172-173: Is the WTIV more conservative, since it is more likely than the BTIV to define a species as Moderate than as Tolerant?

- Lines 175-176: Please add corresponding proportions of fish in each tolerance class for BTIV method.

- Line 222: Please add a single sentence re-stating the main research issue, before summarising the results.

Line 231-232: I still don't understand what 'fish tolerance assessment system we developed'. Have you made novel improvements to Sievert et al. (2016), or is this just a new application of an established method?

- Figure 3b: This is not a very linear relationship. If you separate the BTIV values into a group <0.5 and a group >0.5, you get two classes of sites with almost no positive response to axis 1 score. Figure 3a suggests a more continuous relationship. I wonder if this means that WTIV responds more closely to environmental drivers?

- Figure 3: Should the PCA axis (environmental variables) be the independent (x-axis) variable, as fish species are responding to these drivers - see second aim (Lines 81-82)? 

- Discussion from line 246: Given the differences between WA methods, how do you have confidence that they accurately capture sensitivity?

- Lines 291-293: Can you state which WA method may be better, and give examples of when one method or the other may be more appropriate? Sievert et al. (2016) may have useful insights here?

Author Response

(The authors gave the same response as above.)

Round 2

Reviewer 1 Report

Though I saw some significant improvements over the original version, there are still issues that need to be dealt:

Title: It is inadmissible how, after my recommendation to improve the English, you give such a huge typo on the title: “Study” instead of “Studuy”. Nonetheless, the word “Study” is quite redundant. It’s better something like “Assessment of fish species tolerance….: biological traits vs. weighted averaging approaches”. Once again, I highly recommend the author to have their manuscript revised by a native-English speaker.

Line 34 – Replace “biological assessment índex” by “biological assessment tools”.

Line 90 – Same as in abstract: replace “fish diversity index” by “biological assessment tools”. The same on line 327.

Where are Table 1 and 2? Only Table 3 appear on the manuscript!! Or is this Table 1??

Line 135 – What do these classes mean? Excellent (which one?)? Good (Which one?)?Etc. Provide details

Author Response

Dear reviewer:

Thank you very much for your valuable comments on our manuscript. According to your comments, the manuscript has been revised by a native-English speaker (International Science Editing).

Water EISSN 2073-4441 Published by MDPI AG, Basel, Switzerland RSS E-Mail Table of Contents Alert
Back to Top